# Do Calving-Related Injuries of the Vestibulum Vaginae and the Vagina Affect the Reproductive Performance in Primiparous Dairy Cows?

**DOI:** 10.3390/vetsci10010043

**Published:** 2023-01-07

**Authors:** Helena Marien, Natascha Gundling, Wolfgang Hasseler, Maren Feldmann, Kathrin Herzog, Martina Hoedemaker

**Affiliations:** 1Clinic for Cattle, University of Veterinary Medicine Hannover, 30173 Hannover, Germany; 2Joint Veterinary Practice, 26871 Papenburg, Germany; 3Bovine Health Service Switzerland, 8057 Zürich, Switzerland; 4Department for Animal Welfare Service, Lower Saxony State Office for Consumer Protection and Food Safety, 26203 Oldenburg, Germany

**Keywords:** fertility, heifers, obstetrics, dystocia, metritis

## Abstract

**Simple Summary:**

Dystocia is primarily an issue in heifers. Necessary calving assistance should be available early on and the delivery process should be managed with care in order to avoid injuries to the dam and the neonate. The aim of this study was to investigate the influence of calving-related injuries of the vestibulum vaginae and the vagina on fertility measures in heifers. Results show that, on the one hand, dystocia was a risk factor for injuries of the soft birth canal, and, on the other hand, those injuries were a risk factor for metritis and endometritis. In this study, calving-related injuries of the vestibulum vaginae and the vagina had no statistically significant effect on the reproductive performance of heifers. Nevertheless, our findings emphasized the importance of skilled obstetrical assistance of farmers and veterinarians, especially for heifers, to avoid severe injuries of the soft birth canal and, as a consequence, to prevent a negative impact on animal welfare and the reproductive performance of the animals.

**Abstract:**

The aim of this study was to investigate the influence of calving-related injuries of the vestibulum vaginae and the vagina on fertility measures in heifers. German Holstein heifers (*n* = 236) were checked for vestibulum vaginae and vaginal injuries. These were scored according to localization, depth and length. The healing process was assessed until day 42 post partum. Calving ease and the occurrence of metritis and endometritis were evaluated. In 160 heifers, the following fertility measures were calculated to assess the reproductive performance of heifers: mean interval from calving to first insemination, mean days open, mean interval from first insemination to conception, mean calving interval, mean pregnancy index, percentage of animals pregnant at 200 days p.p., and first service conception rate. On the one hand, dystocia was a risk factor for injuries of the soft birth canal, and, on the other hand, those injuries were a risk factor for metritis and endometritis. In this study, calving-related injuries of the vestibulum vaginae and the vagina had no statistically significant effect on the reproductive performance of heifers. One reason for this outcome was probably the overall good healing tendencies of those injuries in heifers.

## 1. Introduction

The objective of a successful management of cows at calving time is to ensure the delivery of viable calves and smooth transition of the dam from the dry cow period to lactation without complications. Necessary calving assistance should be available early on, and the delivery process should be managed with care in order to avoid injuries to the dam and the neonate. A major problem encountered at calving time is dystocia, which often results in perinatal mortality [1]. The adverse effects of poor calving management are numerous and well documented. Dams with dystocia often have retained fetal membranes and metritis, and, as a consequence, poor fertility measures and impaired milk yield in the following lactation [2,3,4,5]. Dystocia is primarily an issue in heifers [6]. In a study on 203 privately owned cow–calf herds in Canada, where 29,970 full-term births were analyzed, dystocia incidence for primiparous cows was 17.3%, and 2.9% to 4.7% in multiparous cows. Therefore, overall dystocia risk was 8.9% [6]. In another study, 19.0% of heifers and only 6.0% of multiparous females needed assistance during calving [7]. Studies on injuries (Figure 1) occuring after calving in the soft birth canal of cattle are rare. In fact, there is hardly any evidence-based knowledge about the effect of injuries of the soft birth canal on further reproductive performance. A recent study in Germany dealt with injuries after spontaneous parturitions in heifers and cows. Particularly heifers showed significant injuries in the area between the vulva and the hymenal ring. Additionally, good healing tendencies of those injuries was documented [8]. The aim of this study was to investigate the influence of calving-related injuries of the vestibulum vaginae and the vagina on fertility measures in heifers.

## 2. Materials and Methods

The study took place on a large dairy farm in Saxony-Anhalt, Germany. The experimental phase started at the first day post partum (p.p.) and ended at day 200 p.p. Data about the injuries of the vestibulum vaginae and the vagina were collected during an evaluation of an automatic birth monitoring system. In this study, the visual birth monitoring of the farm staff in heifers was complemented by the automated iVET^®^ birth monitoring system. The iVET^®^ consists of two components: a transmitter, which is inserted into the vagina of the animals to be monitored, and a receiver, which must be installed above the calving pen. During the birth process, the transmitter is forced out of the vagina and sends a signal to the receiver, which then triggers an SMS or phone call to the person in charge. Interval from insertion to first birth alarm averaged 74.6 ± 89.2 h [9]. The calving ease was scored (Score 1: spontaneous calving, score 2: slight assistance needed, score 3: severe assistance needed). In 236 heifers (74 animals with visual birth monitoring by farm personnel, 162 with iVet^®^) at the first day p.p., injuries of the vestibulum vaginae (IVV) were examined by splaying the vulva lips manually. Additionally, injuries of the vagina (IV) were examined by manual vaginal palpation at the first day p.p. For the evaluation, all injuries were scored according to localization, depth and length (IVV: degree 0: no injuries; degree 1: marked injuries <2 cm deep; degree 2: severe injuries ≥2 cm deep; IV: degree 0: no injuries, degree 1: lesion <2 cm deep and up to 10 cm long; degree 2: lesion ≥2 cm deep and/or ≥10 cm long). The healing process of the injuries of the vestibulum vaginae was assessed by splaying the vulva lips manually at days 10, 21, and 42 p.p. (degree 1: healing completed (no visible lesions or only scars), degree 2: lesion with good healing tendency (lesion already smaller and in granulation), degree 3: no or poor healing tendency (lesion almost unchanged, often with a lot of necrotic tissue). The healing process of vaginal injuries was not documented, because the owner of the farm did not allow further vaginal examinations. For the evaluation of the association between calving ease, metritis, and endometritis, as well as injuries of the vestibulum vaginae and the vagina, the injuries were categorized into no injuries, mild injuries (maximum injuries of degree 1) and severe injuries (injuries of degree 2, or combinations of no injuries or mild injuries with injuries of degree 2). The occurrence of metritis and endometritis was documented. According to [10] an infection of the uterus within 21 days post partum was classified as metritis. Metritis was characterized by an enlarged uterus and a watery red-brown fluid to viscous purulent uterine discharge, which had a fetid odour. The severity of disease was categorized by the signs of health (metritis degree 1: enlarged uterus and uterine discharge without any systemic signs of illness; metritis degree 2: additional signs of systemic illness, such as decreased milk yield, dullness, and fever >39.5 °C; metritis degree 3: additional signs of toxaemia, such as inappetance, cold extremities, depression, and/or collapse). Clinical endometritis was defined as the presence of a purulent uterine discharge detectable in the vagina 21 days or more post partum. Clinical endometritis was categorized according to the character of the vaginal mucus (endometritis degree 0: clear or translucent mucus; endometritis degree 1 = mucus containing only some flecks of pus; endometritis degree 2 = exudate containing <50% white mucopurulent material; endometritis degree 3 = exudate containing >50% purulent mucus). The following fertility measures were calculated to assess reproductive performance of the heifers: mean interval from calving to first insemination (CFI), mean days open (DO), mean interval from first insemination to conception (FIC), mean calving interval (CI), mean pregnancy index (PI), percentage of animals pregnant at 200 days p.p. (PP), first service conception rate (FCR) [11]. Statistical analyses were carried out using the Statistical Analysis System (Version 9.1, SAS Institute, Cary, NC, USA). Normality of data points was tested using the Shapiro–Wilk test (proc univariate). For calving ease, healing process, occurence of metritis and endometritis, PP, and FCR, a Chi Square or Fisher’s Exact test was used. For CFI, DO, FIC, and CI, Wilcoxon’s two-sample test (PROC NPAR1WAY) was used. Differences were considered significant at *p* ≤ 0.05.

## 3. Results

### 3.1. Occurrence of Injuries of the Vestibulum Vaginae and the Vagina

In 236 heifers, the presence or absence of injuries was evaluated. Only 29 animals did not show any injuries at all. In 207 animals, injuries of the vestibulum vaginae and the vagina were documented. The occurrence of those injuries in the study heifers is presented in Table 1. Most of the animals had injuries of the vestibulum vaginae with or without additional injuries of the vagina.

### 3.2. Healing Process of Injuries of the Vestibulum Vaginae

The healing process of injuries of the vestibulum vaginae was assessed by splaying the vulva lips at days 10, 21, and 42. Degree 1 injuries of the vestibulum vaginae mostly revealed complete healing at day 21 p.p. In contrast, the healing process of injuries of the vestibulum vaginae degree 2 was usually completed by day 42. p.p. (Table 2).

### 3.3. Calving Ease

Spontaneous calving occurred in 31.4% (74 animals) of the heifers. A total of 38.1% (90 animals) of the animals required slight calving assistance and 30.5% (72 animals) required severe calving assistance. Calving ease had a statistically significant effect on the occurrence of injuries of the soft birth canal (*p* < 0.001). The higher the calving ease score, the more severe were the injuries. A total of 79.2% of the animals requiring severe calving displayed severe injuries of the soft birth canal. On the other hand, it is remarkable that 67.6% of the animals with spontaneous calving had mild injuries of the soft birth canal, and 13.5% suffered even severe injuries of the soft birth canal (Table 3).

### 3.4. Association between Injuries of the Soft Birth Canal and Infections of the Uterus

#### 3.4.1. Occurrence of Metritis at Day 10 p.p.

Only 37.4% (86 animals) of the heifers did not display any signs of a metritis at day 10 p.p. With 62.6% (144 animals), the incidence of metritis at day 10 p.p. in the study animals was very high. A statistically significant association between the severity of injuries of the soft birth canal and the occurrence of metritis at day 10 p.p. was found (*p* = 0.0321). The more severe the injuries were, the more likely it was that the animals developed metritis at day 10 p.p. Data of six animals are missing, because the animals died or were culled before day 10 p.p. (Table 4).

#### 3.4.2. Occurrence of Endometritis

A total of 58.7% (132 animals) of the heifers did not display any signs of endometritis at day 21 p.p. With 41.3% (93 animals), the incidence of endometritis at day 21 p.p. was high in the study animals. A statistically significant association between the severity of injuries of the soft birth canal and the occurrence of endometritis at day 21 p.p. was found (*p* < 0.0001). Animals with severe injuries were more likely to develop endometritis at day 21 p.p. than animals with no or mild injuries. Only 25.9% of the animals without injuries displayed signs of endometritis at day 21 p.p. The values of animals with mild injuries were comparable. In contrast, 59.6% of the animals with severe injuries of the soft birth canal showed signs of endometritis at day 21. Data of 11 animals are missing (10 animals were culled before day 21 p.p., the data for one animal are missing) (Table 5).

### 3.5. Reproductive Performance

Seventy-six heifers were not inseminated and left the farm during the study period. A total of 51.3% (39 animals) of those heifers were sold as breeding animals to other farms. A total of 40.8% (31 animals) were culled or died because of a variety of other diseases. A total of 7.8% (six animals = 2.9% of animals with injuries) of those heifers were sold to the slaughterhouse between the 42nd day post partum and the end of the study period because of infertility. One of those six heifers had no injuries andfive had severe injuries of the vagina in combination with marked or severe injuries of the vestibulum vaginae. All of them developed metritis or endometritis post partum. In 160 heifers, fertility measures were calculated to assess reproductive performance. Of those, 64.4% (103 animals) became pregnant. Compared with target values, the insemination success of all heifers was low, but the target values concerning the timeline parameters were met [11]. There was no statistically significant difference between the fertility measures of animals with or without injuries (Table 6).

To assess the effect of the severity of the injuries of the vestibulum vaginae on the reproductive performance of the heifers, the fertility measures were evaluated according to the degree of the injuries and location in detail. Most of the comparisons between the subgroups showed no statistically significant differences. However, one aspect is worth mentioning in detail. Fifty-one animals showed mild injuries of the vestibulum vaginae only. Compared with the target values, their reproductive performance was satisfactory. Two heifers had severe injuries of the vestibulum vaginae only. Those two animals displayed a statistically significant longer mean interval from calving to first insemination compared with animals without injuries or with mild injuries of the vestibulum vaginae It was remarkable that none of those animals became pregnant.

### 3.6. Association between Infections of the Uterus and Reproductive Performance

In 159 animals, infections of the uterus or the absence of signs of infections were documented. Data of one animal are missing. No statistically significant association between the occurrence of metritis at day 10 p.p. or endometritis at day 21 p.p. and the reproductive performance of the study heifers was found. Compared with target values, the insemination success of all groups was low. The mean interval from calving to conception was relatively high. In the other timeline parameters, the target values were met in all groups (Table 7).

## 4. Discussion

The aim of this study was to investigate the influence of calving-related injuries of the vestibulum vaginae and the vagina on fertility in heifers.

### 4.1. Injuries of the Vestibulum Vaginae and the Vagina

Almost 90% of the study heifers displayed injuries of the soft birth canal tissue at the first day after calving. In the study of [8], particularly, heifers also had injuries of the soft birth canal tissue. Half of the injuries in our study were mild and the other half were severe. While there is a lot of literature describing the involution of the uterus after calving, there is little information available regarding possible alterations in the soft birth canal after calving in cattle. In a study from 2000 in a large dairy herd (1420 animals, Holstein–Friesean), most of the documented injuries were pneumovagina, and the highest incidence thereof was found in heifers. In this study, heifers had a six times higher risk of injuries than other parities [12]. This corresponds to our findings, as almost 80% of our study animals had injuries of the vestibulum vaginae, and a pneumovagina is normally the consequence of a deficient closing of the vulva lips caused by scars from injuries of the vestibulum vaginae.

### 4.2. Healing Process of Injuries of the Vestibulum Vaginae

In general, injuries of the vestibulum vaginae had a very good healing tendency. Mild injuries of the vestibulum vaginae mostly revealed complete healing by day 21 p.p. The healing process of severe injuries of the vestibulum vaginae was completed by day 42 p.p. This corresponds with the findings of [8]. The authors, who examined the injury pattern of the soft birth canal in heifers and pluriparous cows after spontaneous calvings in detail, also emphasized the good healing tendency of the injuries.

### 4.3. Calving Ease

Spontaneous calving occurred in 31.4% of the heifers. A total of 38.1% of the animals required slight calving assistance and 30.5% required severe calving assistance. Calving ease had a statistically significant effect on the occurrence of injuries of the soft birth canal in our study. The higher the calving ease score, the more severe were the injuries. Our evaluation of the calving ease showed that 79.2% of the animals with a calving ease score 3 showed severe injuries of the soft birth canal. Additionally, animals with a retention time of the iVet^®^-sensor of more than 24 h needed severe calving assistance statistically significantly more often [13]. To our knowledge, to date, only a few studies have dealt with injuries after dystocia in heifers in detail, even though it is well known that heifers often require assistance at calving [8]. Especially in heifers, premature obstetrical assistance leads to a high prevalence of dystocia [1]. When the obstetrical assistance is not managed with patience and care, the tissue ruptures because there is insufficient time for widening of the soft birth canal to occur. Additionally, it is known that, in heifers, periparturient environmental stress is associated with an incomplete dilatation of the vulva or cervix [14,15]. Perhaps this could increase the occurrence of injuries of the soft birth canal tissue, too. On the other hand, it is remarkable that 67.6% of the animals with spontaneous calving had mild injuries of the soft birth canal and 13.5% even suffered severe injuries of the soft birth canal. This also corresponds with the findings of [8], where mild injuries of the soft birth canal tissue after spontaneous calvings were documented in all of the 25 study heifers and all of the 25 study cows.

### 4.4. Association between Injuries of the Soft Birth Canal and Infections of the Uterus

The incidence of metritis at day 10 p.p. (62.6%) and the incidence of endometritis at day 21 p.p. (41.3%) in the study animals were very high. A statistically significant association between the severity of injuries of the soft birth canal and the occurrence of metritis at day 10 p.p. and endometritis at day 21 p.p. was found. The more severe the injuries were, the more likely it was that the animals had an infection of the uterus post partum. According to the literature, three common reasons for disturbed health after assisted calvings are: a higher prevalence of retained placenta, endometritis, and/or vulvovaginal lacerations compared with unassisted calvings [16,17]. Therefore, one reason for this outcome may be the high percentage of assisted calvings in the study heifers and the high percentage of severe injuries of the soft birth canal tissue. Additionally, the iVet^®^-sensor was an important risk factor for infections of the uterus, because animals with a retention time of the sensor of more than 24 h developed significantly more often retained fetal membranes, metritis at day 10 p.p., and endometritis at day 21 p.p. [13].

### 4.5. Association between Injuries of the Soft Birth Canal and the Reproductive Performance

The main topic of our study was to evaluate the influence of injuries of the soft birth canal tissue on fertility measures in heifers. Compared with target values, the insemination success of all study heifers was low, but the target values concerning the timeline parameters were met [11]. In general, there was no statistically significant difference between the fertility measures of animals with or without injuries. However, our study allowed no further evaluation of the fertility parameters concerning the effect of the severity and localization of the injuries, because of the low number of animals in those subgroups. However, it was remarkable that the interval between calving to first insemination in animals with severe injuries of the vestibulum vaginae was statistically significantly higher than in animals without or with mild injuries. None of those animals became pregnant. Additionally, it has to be mentioned that 2.9% (*n* = 6) of the heifers were culled because of infertility, and, therefore, received no insemination. Five of those animals suffered from severe injuries and developed a metritis or endometritis. In recent studies, there was no statistically significant effect of mild dystocia on the reproductive performance of dairy cows. Those cows with severe dystocia, especially if a Cesarian section was necessary, had impaired fertility [18,19,20]. As mentioned before in our study, we did not find a distinct negative effect of the injuries on the fertility measures. Most likely, one reason is the very good healing tendency of those injuries. In that way, the injuries showed a completed healing several weeks before the start of the insemination period. One explanation for the low insemination success compared with target values was probably the high incidence of infections of the uterus in the study heifers. The negative effects of clinical and subclinical endometritis (for example, prolonged mean interval from calving to first insemination and low conception rate) are well documented [21,22,23]. However, in our study, we did not find a negative effect of infections of the uterus on fertility measures in heifers. Perhaps one explanation is that our study is dealt solely with heifers, which have fewer additional health problems (for example ketosis or hypocalcaemia) than pluriparous animals [24]. Ketosis impairs the immune defence system of the animals and the risk of infections of the reproductive tract or the udder is higher than in healthy animals [3,5,20]. This could have supported a quick healing process of the infections of the uterus post partum. Maybe the well known negative effect of infections of the uterus on the reproductive performance in cattle is mitigated in our study heifers in that way. Independent of the occurrence of injuries of the soft birth canal or infections of the uterus, the reproductive performance of the study heifers was not satisfactory. Perhaps other common reasons for an impaired fertility at the herd level, such as insufficient heat detection management, often in combination with a high lameness prevalence [25,26], or long periods of negative energy balance [27], played a role in this respect. Nevertheless, our findings emphasized the importance of skilled obstetrical assistance of farmers and veterinarians, especially for heifers, to avoid severe injuries of the soft birth canal and, as a consequence, to prevent a negative impact on animal welfare and the reproductive performance of the animals.

## 5. Conclusions

In this study, calving-related injuries of the vestibulum vaginae and the vagina had no statistically significant effect on the reproductive performance of heifers. One reason for this outcome was probably the overall good healing tendency of those injuries in heifers.

## Figures and Tables

**Figure 1 vetsci-10-00043-f001:**
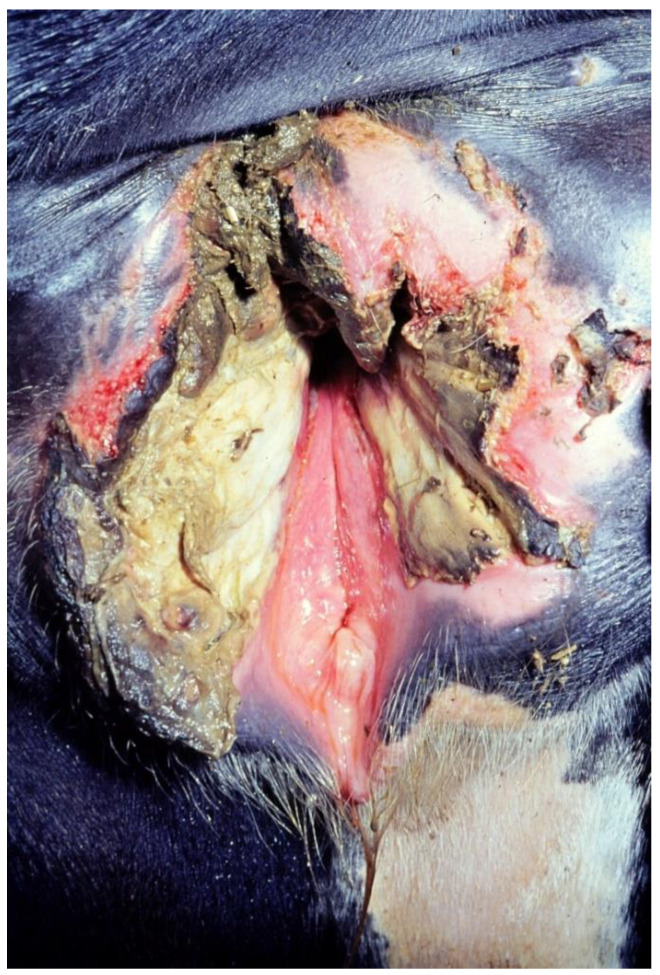
Injury of the vestibulum vaginae after dystocia in a heifer (picture source: Clinic for Cattle, University of Veterinary Medicine Hannover).

**Table 1 vetsci-10-00043-t001:** Occurrence of the injuries of the vestibulum vaginae and vaginae (*n* = 236). Degrees of injuries of the vestibulum vaginae: degree 0: no injuries; degree 1: marked injuries <2 cm deep; degree 2: severe injuries ≥2 cm deep; degrees of injuries of the vagina: degree 0: no injuries, degree 1: lesion <2 cm deep and up to 10 cm long; degree 2: lesion ≥2 cm deep and/or ≥10 cm long (*n* = number of animals).

Degree of Injuries	*n*	%
No injuries (0)	29	12.3
Vestibulum 1 + Vagina 0	69	29.3
Vestibulum 2 + Vagina 0	4	1.7
Vestibulum 0 + Vagina 1	13	5.5
Vestibulum 0 + Vagina 2	11	4.7
Vestibulum 1 + Vagina 1	30	12.7
Vestibulum 1 + Vagina 2	62	26.3
Vestibulum 2 + Vagina 1	4	1.7
Vestibulum 2 + Vagina 2	14	5.9
Total	236	100.1 *

* due to rounding.

**Table 2 vetsci-10-00043-t002:** Healing process of the mild and severe injuries in the vestibulum vaginae. Degrees of injuries of the vestibulum vaginae: degree 1: marked injuries <2 cm deep; degree 2: severe injuries ≥2 cm deep.

	Injuries of the Vestibulum Vaginae	
Healing Completed	Degree 1 (*n*)	Degree 2 (*n*)
Day 10 postpartum (%) Day 21 postpartum (%) Day 42 postpartum (%)	19.2 (31) 77.8 (125) ^a^ 2.9 (5) ^a^	4.5 (1) 40.9 (9) ^b^ 54.5 (12) ^b^
Total (183)	161	22

^a,b^ values with varying letter superscripts within a row are statistically significantly different.

**Table 3 vetsci-10-00043-t003:** Association between calving ease (score 1: spontaneous calving, score 2: slight assistance needed, score 3: severe assistance needed) and the occurrence of injuries of the vestibulum vaginae and/or the vagina (no injuries, mild injuries: maximum injuries of degree 1, severe injuries: injuries of degree 2, or combinations of no injuries or injuries of degree 1 with injuries of degree 2).

Calving Ease	No Injuries (*n*)	Mild Injuries (*n*)	Severe Injuries (*n*)	*p*-Value
Score 1 (%) Score 2 (%) Score 3 (%)	18.9 (14) 16.7 (15) 0.0 (0)	67.6 (50) 52.2 (47) 20.8 (15)	13.5 (10) 31.1 (28) 79.2 (57)	<0.001
Total (236)	29	112	95	

**Table 4 vetsci-10-00043-t004:** Association between injuries of the soft birth canal (no injuries, mild injuries: maximum injuries of degree 1, severe injuries: injuries of degree 2, or combinations of no injuries or injuries of degree 1 with injuries of degree 2) and the occurrence of metritis at day 10 p.p.

	No Injuries (*n*)	Mild Injuries (*n*)	Severe Injuries (*n*)	*p*-Value
No metritis (%) Metritis at day 10 p.p. (%)	51.7 (15) 48.3 (14)	41.4 (46) 58.6 (65)	27.8 (25) 72.2 (65)	0.0321
Total (230)	29	111	90	

**Table 5 vetsci-10-00043-t005:** Association between injuries of the soft birth canal (no injuries, mild injuries: maximum injuries of degree 1, severe injuries: injuries of degree 2, or combinations of no injuries or injuries of degree 1 with injuries of degree 2) and the occurrence of endometritis at day 21 p.p.

	No Injuries (*n*)	Mild Injuries (*n*)	Severe Injuries *(n*)	*p*-Value
No endometritis at day 21 p.p. (%) Endometritis at day 21 p.p. (%)	74.1 (20) 25.9 (7)	69.7 (76) 30.3 (33)	40.5 (36) 59.6 (53)	<0.0001
Total (225)	27	109	89	

**Table 6 vetsci-10-00043-t006:** Fertility measures in heifers with and without injuries of the vestibulum vaginae and/or the vagina (SD = standard deviation).

Fertility Measures	No Injuries (SD)	Injuries (SD)	Target ^1^
	*n* = 19	*n* = 141	
Mean interval from calving to first insemination (d)	93.7 (±19.1)	97.3 (±25.7)	≤85–90
Mean days open (d)	111.7 (±38.3)	116.7 (±38.1)	≤115
Mean interval from first insemination to conception (d)	26.3 (±33.2)	22.4 (±28.9)	≤18
Mean calving interval (d)	391.7 (±38.3)	396.7 (±38.1)	≤400
Mean pregnancy index	1.8 (±0.8)	1.8 (±0.8)	≤1.7
Percentage of animals pregnant at 200 days p.p. (%)	52.6	67.6	≥90
First service conception rate (%)	22.2	33.9	≥50

^1^ [11].

**Table 7 vetsci-10-00043-t007:** Fertility measures in heifers with or without metritis at day 10 p.p. or endometritis at day 21 post partum (SD = standard deviation, p.p. = post partum).

Fertility Measures	No Metritis at Day 10 p.p. (SD) *n* = 63	Metritis at Day 10 p.p. (SD) *n* = 96	No Endometritis at Day 21 p.p. (SD) *n* = 101	Endometritis at Day 21 p.p. (SD) *n* = 58
Mean interval from calving to first insemination (d)	96.4 (±24.1)	99.3 (±25.4)	97.2 (±23.8)	99.7 (±26.7)
Mean days open (d)	109.0 (±34.3)	118.3 (±41.0)	114.8 (±39.9)	114.1 (±36.6)
Mean interval from first insemination to conception (d)	18.4 (±23.4)	24.3 (±33.4)	22.7 (±29.7)	20.5 (±30.1)
Mean calving interval (d)	389.0 (±34.3)	400.0 (±41.6)	396.7 (±40.7)	394.2 (±36.6)
Mean pregnancy index	1.6 (±0.6)	1.7 (±1.1)	1.6 (±0.7)	1.8 (±1.2)
% animals pregnant at 200 days p.p. (%)	66.7	63.5	63.4	67.2
First service conception rate (%)	32.8	37.9	31.6	44.7

## Data Availability

Data are contained within the article.

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
