# Peer review of "Do Calving-Related Injuries of the Vestibulum Vaginae and the Vagina Affect the Reproductive Performance in Primiparous Dairy Cows?"

_vetsci, 2023, doi:10.3390/vetsci10010043_

Round 1

Reviewer 1 Report

 This Interest study that explain the influence of lesion of calving in the posterior reproduccion performance .

May be nec esary add son images or pictures about vaginal and vestibular lesion

Author Response

Dear Reviewer 1,

thank you very much for your friendly comments. I have inserted a picture of an injury of the vestibulum vaginae.

Kind regards

Natascha Gundling

Reviewer 2 Report

This study sought to determine the effects of the location and severity of vaginal or vaginal vestibular injury associated with first calving in heifers on subsequent reproductive performance. The results that calving ease had a statistically significant effect on the occurrence of soft birth canal injury and that injury was more severe with higher calving ease scores are consistent with most previous studies and are not novel. On the other hand, it is interesting to note that while the prevailing theory has been that birth canal injury has a negative impact on subsequent fertility, the data in this study show that it did not have a statistically significant negative impact on reproductive performance, which is a result that overturns the prevailing theory. 

However, it may be difficult to understand how one could conclude that soft birth canal injury is not associated with subsequent reproductive performance, despite the statistically significant association between the severity of soft birth canal injury and the occurrence of uterine inflammation and endometritis. This discrepancy needs to be explained because it is well-known that postpartum uterine disease has a negative effect on subsequent reproductive performance.

This reviewer is concerned that the number of the animals used in this study is too small to compare reproductive performance. In this sense, Tables 7, 8, and 9 should be deleted.

Another point of major concern is that the authors state in their discussion that "one of the causes of the injury of the soft birth canal tissue in this study may be the intravaginal iVet® sensor itself." If this device is the cause, then the results of this study mislead the readers.

Furthermore, the statement at the end of the abstract and at the end of the conclusion of the text should be reconsidered because it is difficult to understand. Is it that the healing of the injuries was very good because the subject cows happened to be well managed on the farm? Or do the authors try to imply that birth canal injuries caused by dystocia in general tend to heal spontaneously by the subsequent breeding time? In addition, it is not clear about the description in abstract that because only heifers were used in the study, there was no significant difference in the effect of the birth canal injury on reproductive performance.

Other specific comments are noted below.

L64-66: Regarding the iVet® sensor, the authors should not only cite the literature, but also describe in more detail when it was applied and how it was used.

L75-78: The authors mention that the healing process of the injuries of the vestibulum vaginae was assessed by splaying the vulva lips manually, but they should specifically describe the definition of completed healing and the differences between lesions with good healing tendency and those with poor healing tendency (e.g., necrotic tissue, pus discharge, granulation, wound shrinkage, scar formation, etc.).

L83-84: The definitions of metritis and endometritis should not only be cited in the literature, but should also be stated in terms of how many days postpartum the diagnosis was made and what findings were used as diagnostic criteria in this study.

L159-170: Considering that some of the 76 heifers that were not used to calculate reproductive performance were culled, died, or sold because they could not be inseminated due to birth canal injury, would it be possible that birth canal injury at calving in heifers impairs reproductive performance? This point needs to be mentioned in the discussion.

L184-186, 196-197: The statement that the small number of heifers should be taken into account when interpreting the results of statistical analysis should not be stated in the "Results" but in the "Discussion" section.

L210-211: Since the analysis was performed on a small number of animals, it is not possible to make the conclusion as in this sentence, is it?

L238-240: Results that are not described in the "Results" section should not be described in “Discussion” section. If the authors intend to describe the relationship between intravaginal retention time of iVet®-sensor and intravaginal injury, they should add it in the "Results" section. The sentences in L280-283 would then become clear in relation to the results.

L243-244: If the injuries are caused by the intravaginal iVet®-sensor, wouldn't it be natural that they would be minor and heal quickly?

Author Response

Response to Reviewer 2 comments

Thank you very much for your percise and constructive comments.

This study sought to determine the effects of the location and severity of vaginal or vaginal vestibular injury associated with first calving in heifers on subsequent reproductive performance. The results that calving ease had a statistically significant effect on the occurrence of soft birth canal injury and that injury was more severe with higher calving ease scores are consistent with most previous studies and are not novel. On the other hand, it is interesting to note that while the prevailing theory has been that birth canal injury has a negative impact on subsequent fertility, the data in this study show that it did not have a statistically significant negative impact on reproductive performance, which is a result that overturns the prevailing theory. 

However, it may be difficult to understand how one could conclude that soft birth canal injury is not associated with subsequent reproductive performance, despite the statistically significant association between the severity of soft birth canal injury and the occurrence of uterine inflammation and endometritis. This discrepancy needs to be explained because it is well-known that postpartum uterine disease has a negative effect on subsequent reproductive performance.

=> This result was a big surprise for us, too. I have tried to explain this issue more precise in the discussion.

This reviewer is concerned that the number of the animals used in this study is too small to compare reproductive performance. In this sense, Tables 7, 8, and 9 should be deleted.

=> You are right. I have deleted Tables 7, 8 and 9.

Another point of major concern is that the authors state in their discussion that "one of the causes of the injury of the soft birth canal tissue in this study may be the intravaginal iVet® sensor itself." If this device is the cause, then the results of this study mislead the readers.

=> The sensor is certainly not the main reason for the injuries, because other studies showed that mild injuries of the soft birth canal tissue are very common after calving. Therefore, I deleted the remarks about the sensor in this part of the article.

Furthermore, the statement at the end of the abstract and at the end of the conclusion of the text should be reconsidered because it is difficult to understand. Is it that the healing of the injuries was very good because the subject cows happened to be well managed on the farm? Or do the authors try to imply that birth canal injuries caused by dystocia in general tend to heal spontaneously by the subsequent breeding time? In addition, it is not clear about the description in abstract that because only heifers were used in the study, there was no significant difference in the effect of the birth canal injury on reproductive performance.

=> You are right. I have corrected the conclusion.

Other specific comments are noted below.

L64-66: Regarding the iVet® sensor, the authors should not only cite the literature, but also describe in more detail when it was applied and how it was used.

=> The describtion of the sensor was complemented to material and methods.

L75-78: The authors mention that the healing process of the injuries of the vestibulum vaginae was assessed by splaying the vulva lips manually, but they should specifically describe the definition of completed healing and the differences between lesions with good healing tendency and those with poor healing tendency (e.g., necrotic tissue, pus discharge, granulation, wound shrinkage, scar formation, etc.).

=> Definition of the healing process was complemented (materials and methods).

L83-84: The definitions of metritis and endometritis should not only be cited in the literature, but should also be stated in terms of how many days postpartum the diagnosis was made and what findings were used as diagnostic criteria in this study.

=> Definitions of metritis and endometritis were complemented (materials and methods).

L159-170: Considering that some of the 76 heifers that were not used to calculate reproductive performance were culled, died, or sold because they could not be inseminated due to birth canal injury, would it be possible that birth canal injury at calving in heifers impairs reproductive performance? This point needs to be mentioned in the discussion.

=> You are right. I explained this issue more precise in the discussion.

L184-186, 196-197: The statement that the small number of heifers should be taken into account when interpreting the results of statistical analysis should not be stated in the "Results" but in the "Discussion" section.

=> You are right. I corrected this statement.

L210-211: Since the analysis was performed on a small number of animals, it is not possible to make the conclusion as in this sentence, is it?

=> You are right. I have deleted this sentence.

L238-240: Results that are not described in the "Results" section should not be described in “Discussion” section. If the authors intend to describe the relationship between intravaginal retention time of iVet®-sensor and intravaginal injury, they should add it in the "Results" section. The sentences in L280-283 would then become clear in relation to the results.

=> You are right. I have corrected this issue in the discussion.

L243-244: If the injuries are caused by the intravaginal iVet®-sensor, wouldn't it be natural that they would be minor and heal quickly?

=> Yes. You are right.

Reviewer 3 Report

In this study, Marien et al examine the influence of calving related injuries of the vestibulum vaginae and the vagina on fertility measures in heifers. The study is interesting and is of broad interest. Prior to acceptance of this manuscript for publication, the following comments needs to be addressed.

1. The paper could benefit substantially from proof-reading and editing. There are several instances where syntax and grammar are not proper. This needs to be rectified.

2. Lines 45, 77, 79 etc- unnecessary use of hyphens

3. When classification of injuries were done, why was length not a factor for classification of injury in the vestibulum vagina but used only on the vagina? If it is due to difficulty in measuring, how was depth measured?

4. Definition of metritis and endometritis needs to be elaborated in the paper as these are parameters being examined. In the present form, there is just a reference to an article.

5. How was puerperal metritis classified in the scheme? Was it along with metritis?

According to the paper cited in this manuscript, "Puerperal metritis should be defined as an animal with an abnormally enlarged uterus and a fetid watery red-brown uterine discharge, associated with signs of systemic illness (decreased milk yield, dullness or other signs of toxemia) and fever > 39.5 degrees C, within 21 days after parturition.

Was systemic illness factored in these animals when considering classification?

Was subclinical endometritis given consideration?

Author Response

Response to Reviewer 3 comments

Thank you very much for your comments.

1. The paper could benefit substantially from proof-reading and editing. There are several instances where syntax and grammar are not proper. This needs to be rectified.

=> MDPI was asked for English editing.

2. Lines 45, 77, 79 etc- unnecessary use of hyphens

=> corrected

3. When classification of injuries were done, why was length not a factor for classification of injury in the vestibulum vaginae but used only on the vagina? If it is due to difficulty in measuring, how was depth measured?

=> You are right. We should have measured the length of the injuries of the vestibulum vagniae, but unfortunately we did not do this.

4. Definition of metritis and endometritis needs to be elaborated in the paper as these are parameters being examined. In the present form, there is just a reference to an article.

=> I added the classification of metritis and endometritis in the manuscript.

5. How was puerperal metritis classified in the scheme? Was it along with metritis?

=> Puerperal metritis was along with metritis degree 2.

According to the paper cited in this manuscript, "Puerperal metritis should be defined as an animal with an abnormally enlarged uterus and a fetid watery red-brown uterine discharge, associated with signs of systemic illness (decreased milk yield, dullness or other signs of toxemia) and fever > 39.5 degrees C, within 21 days after parturition. Was systemic illness factored in these animals when considering classification?

=> I’m very sorry, because I cited the wrong article of Sheldon et al.. We classified metritis and endometritis according to Sheldon et al. (2009).  Systemic illness was factored in this classification. I have corrected the citation in the manuscript and I added the classification of metritis and endometritis in the manuscript.

Was subclinical endometritis given consideration?

=> We did not consider subclinical endometritis in our study.

Round 2

Reviewer 3 Report

Authors have addressed all comments and concerns.